# White blood cell count and incidence of hypertension in the general Japanese population: ISSA-CKD study

**Shintaro Ishida**[1]*, **Seiji Kondo**[1], **Shunsuke Funakoshi**[2], **Atsushi Satoh**[2], **Toshiki Maeda**[2], **Miki Kawazoe**[2], **Chikara Yoshimura**[2], **Kazuhiro Tada**[3], **Koji Takahashi**[3], **Kenji Ito**[3], **Tetsuhiko Yasuno**[3], **Kosuke Masutani**[3], **Hitoshi Nakashima**[3], **Hisatomi Arima**[2]

1 Department of Oral and Maxillofacial Surgery, Faculty of Medicine, Fukuoka University, Fukuoka, Japan, 2 Department of Preventive Medicine and Public Health, Faculty of Medicine, Fukuoka University, Fukuoka, Japan, 3 Division of Nephrology and Rheumatology, Department of Internal Medicine, Faculty of Medicine, Fukuoka University, Fukuoka, Japan

* shin0723@adm.fukuoka-u.ac.jp

**Data Availability Statement:** All relevant data are within the manuscript and its Supporting Information files.

## Abstract

### Objectives

This study aimed to clarify the relationship between the white blood cell (WBC) count and hypertension in the general Japanese population.

### Methods

We conducted a population-based retrospective cohort study using annual health check-up data of residents of Iki City, Nagasaki Prefecture, Japan. A total of 2935 participants without hypertension at baseline were included in the present analysis. WBC counts were classified as tertile 1 (<4700/μL), tertile 2 (4700–5999/μL), and tertile 3 (≥6000/μL). The outcome was incident hypertension (blood pressure ≥140 mmHg). Multivariable-adjusted hazard ratios and 95% confidence intervals (95% CIs) were estimated using the Cox proportional hazards model.

### Result

During an average follow-up of 4.5 years, 908 participants developed hypertension. The incidence (per 100 person-years) of hypertension increased with an elevation in the WBC count (6.3 in tertile 1, 7.0 in tertile 2, and 7.4 in tertile 3). This association was significant, even after adjustment for other risk factors, including age, sex, current smoking habits, current alcohol intake, exercise habits, obesity, elevated blood pressure, diabetes mellitus, and dyslipidemia. The hazard ratios were 1.07 for tertile 2 (95% CI 0.90–1.26) and 1.27 for tertile 3 (95% CI 1.06–1.51) compared with the reference group of tertile 1 (p = 0.009).

### Conclusion

The WBC count was associated with future development of hypertension in the general Japanese population.

**Funding:** This study was supported by research grants from Iki City (Grant numbers 180424, 190595 and 200492; recipients HA, HN and KM).

**Competing interests:** The authors have read the journal's policy, and the authors of this study have the following competing interests to declare: HA received funding from Daiichi Sankyo and Takeda, lecture fees from Bayer, Daiichi Sankyo, Fukuda Denshi, MSD, Takeda and Teijin, and fees for consultancy from Kyowa Kirin outside of this work. This does not alter our adherence to PLOS ONE policies on sharing data and materials. There are no patents, products in development or marketed products associated with this research to declare. There are no other conflicts of interest to disclose.

## Introduction

Cardiovascular disease is a leading cause of premature death in Japan, as well as in other countries in the world [1, 2]. Approximately 50% of deaths from cerebrovascular disease and 59% from coronary artery disease are attributable to hypertension [2, 3]. In Japan, the average blood pressure (BP) value of the population has decreased, which is mainly due to an increase in the number of people who have received BP-lowering medication and improved management [2–5]. However, the prevalence of hypertension has not decreased during the past few decades [2–5]. Effective prevention of hypertension and subsequent cardiovascular disease requires strategies that are based on current knowledge of risk factors in Japan.

Chronic inflammation is a risk factor of hypertension [1, 2, 4–9]. However current evidence of this risk factor is mainly based on the association between high-sensitivity C-reactive protein and hypertension, and is mainly derived from Western populations [10–13]. Therefore, whether the white blood cell (WBC) count can also predict development of hypertension in the Asian population is unclear [14]. Therefore, this study aimed to clarify the relationship between the WBC count and hypertension in the general Japanese population.

## Subjects and methods

### Study design and participants

We used data from the Iki City Epidemiological Study of Atherosclerosis And Chronic Kidney Disease (ISSA-CKD), which is a population-based retrospective cohort study of the residents of Iki City, Nagasaki Prefecture, Japan. Details of the ISSA-CKD study have been described previously [15]. Iki City consists of a group of islands that are located in the north of Nagasaki Prefecture. The total population of Iki City is approximately 27,000.

Between 2008 and 2017, a total of 7895 residents aged 30 years or older underwent annual health checks that were conducted by the local government of Iki City. We excluded 1881 residents with a follow-up duration of <1 year, 2812 who had hypertension (BP ≥140/90 mm Hg or use of BP-lowering medications) at baseline, and 267 with missing information on the WBC count at baseline. Finally, a total of 2935 participants were included in the present analysis. This study was approved by the Fukuoka University Clinical Research and Ethics Centre (No. 2017M010).

### Data collection

At each health check-up, height and weight were measured with the participant wearing light clothes without shoes, and body mass index (BMI, kg/m$^2$) was calculated. Obesity was defined as a BMI ≥25 kg/m$^2$ [16]. BP was measured by trained staff in the right upper arm using mercury, automated, or aneroid sphygmomanometers with appropriately-sized cuffs, after at least 5 min of rest in a sitting position, in accordance with standardized guidelines [17]. BP was measured twice and the mean of the two values was used in the present analysis. Elevated BP was defined as a BP level of 130–139/80–89 mmHg [2].

Casual blood and urine samples were collected. The WBC count was determined using a multi-item automatic hemocytometer (XN-1000®; Sysmex Corporation, Kobe, Japan). Participants were classified into tertile groups of the WBC count as follows: tertile 1 (<4700/μL), tertile 2 (4700–5999/μL), and tertile 3 (≥6000/μL). Plasma glucose concentrations were determined using an enzymatic method and glycated hemoglobin (HbA1c) levels (NGSP value) were determined using high-performance liquid chromatography. The presence of diabetes was defined by a fasting glucose concentration ≥7.0 mmol/L, a non-fasting glucose concentration ≥11.1 mmol/L, HbA1c value ≥6.5% [18], or the use of glucose-lowering therapies.

Serum low-density lipoprotein (LDL) cholesterol, high-density lipoprotein (HDL) cholesterol, and triglyceride concentrations were determined enzymatically. Dyslipidemia was defined by LDL cholesterol concentrations ≥3.62 mmol/l, HDL cholesterol concentrations <1.03 mmol/l, triglyceride concentrations ≥1.69 mmol/l [2], or the use of lipid-lowering medication. Information regarding the participants' smoking habits, alcohol intake, and regular exercise was obtained using a standard questionnaire. Current smokers were defined as participants who had smoked 100 cigarettes or more, or those who had smoked regularly for more than 6 months at baseline. Alcohol intake was classified into current daily drinking or no daily drinking. Regular exercise was defined as exercise habits of ≥30 minutes per day, two times or more per week.

### Definition of outcome

The outcome of the present analysis was development of hypertension (BP ≥140 mmHg or initiation of BP-lowering medications), which was confirmed at the end of follow-up.

### Statistical analysis

Continuous variables are expressed as the mean ± SD, and trends across tertile groups of the WBC count were tested using simple regression models. Categorical variables are expressed as the number of participants (percentage), and trends across groups were tested using logistic regression models. Incidence rates of hypertension were calculated using the person-year approach. Crude and multivariable-adjusted hazard ratios (HRs) and their 95% confidence intervals (95% CIs) were estimated using the Cox proportional hazards model. The effects of the WBC count on development of hypertension were compared between subgroups defined by other risk factors (age, sex, smoking, obesity, and dyslipidemia) by adding an interaction term to the statistical models. A two-tailed p value of <0.05 was considered statistically significant. All data analyses were carried out using SAS version 9.4.

## Results

Table 1 shows the baseline characteristics according to tertile groups of the WBC count. Participants with a higher WBC count were younger and more likely to be men, with higher rates of current smoking, current daily alcohol intake, obesity, and dyslipidemia.

During an average follow-up of 4.5 years, 908 participants developed hypertension. Table 2 shows the risks of hypertension according to tertile groups of the WBC count. The incidence rate of hypertension increased with elevation of the WBC count (6.3% per 100 person-years in tertile 1, 7.0% in tertile 2, and 7.4% in tertile 3). This association was significant after adjustment for other risk factors, including age, sex, current smoking habits, current alcohol intake, exercise habits, obesity, elevated BP, diabetes, and dyslipidemia (HRs of 1.07 [95% CI 0.90–1.26] for tertile 2 and 1.27 [1.06–1.51] for tertile 3 compared with tertile1 (p = 0.009).

Table 3 shows multivariable-adjusted HRs of the WBC count for the incidence of hypertension in subgroups. There were no clear differences in the effects of the WBC count on hypertension in subgroups defined by age (<65 vs. ≥65 years), sex, obesity, smoking, and dyslipidemia (all p>0.1 for interactions).

## Discussion

The present observational study of the general Japanese population showed a close association between the WBC count and future development of hypertension. This association was significant after adjustment for the effects of confounding factors, such as age, sex, current smoking

**Table 1. Baseline characteristics according to tertiles of the white blood cell count.**

| | White blood cell count (/μL) | | | p value for trend |
|---|---|---|---|---|
| | <4,700/μL (N = 951) | 4,700–5,999/μL (N = 1040) | ≥6,000/μL (N = 944) | |
| Age, mean (SD), years | 59.0 (10.7) | 57.5 (11.4) | 54.3 (12.1) | <0.0001 |
| Male, N (%) | 273 (28.7) | 472 (45.4) | 505 (53.5) | <0.0001 |
| Smoking, N (%) | 81 (8.5) | 157 (15.1) | 361 (38.2) | <0.0001 |
| Current daily alcohol intake, N (%) | 127 (13.5) | 218 (21.2) | 230 (24.7) | <0.0001 |
| Regular exercise*, N (%) | 273 (25.5) | 252 (24.6) | 214 (23.1) | 0.2290 |
| Body mass index, mean (SD), kg/m$^2$ | 22.4 (3.1) | 23.0 (3.2) | 23.5 (3.3) | <0.0001 |
| Obesity**, N(%) | 168 (17.7) | 250 (24.0) | 274 (29.0) | <0.0001 |
| Systolic blood pressure, mean (SD), mmHg | 118.4 (12.1) | 119.2 (12.1) | 118.8 (11.9) | 0.4983 |
| Diastolic blood pressure, mesa (SD), mmHg | 69.7 (8.6) | 70.9 (8.2) | 70.8 (8.8) | 0.0046 |
| Elevated blood pressure***, N (%) | 292 (30.7) | 368 (35.4) | 303 (32.1) | 0.5154 |
| Diabetes mellitus****, N (%) | 37 (3.8) | 53 (5.1) | 55 (5.8) | 0.0525 |
| HbA1c, mean (SD) | 5.1 (0.5) | 5.1 (0.7) | 5.2 (0.8) | 0.0128 |
| High-density lipoprotein cholesterol, mean (SD), mmol/L | 1.70 (0.42) | 1.63 (0.44) | 1.55 (0.40) | <0.0001 |
| Low-density lipoprotein cholesterol, mean (SD), mmol/L | 3.14 (0.80) | 3.19 (0.82) | 3.23 (0.84) | 0.0251 |
| Triglyceride, mean (SD), mmol/L | 1.08 (0.76) | 1.25 (0.31) | 1.46 (1.12) | <0.0001 |
| Dyslipidemia*****, mean (SD), mmol/L | 332 (34.1) | 250 (24.0) | 274 (29.0) | <0.0001 |

*Exercise habits of ≥30 minutes per day for twice or more per week.

** Body mass index of ≥25 kg/m$^2$.

*** BP of 130–139/80–89 mmHg. Body mass index of ≥25 kg/m$^2$.

****Casual serum glucose concentrations ≥11.1 mmol/L, HbA1c values ≥6.5%, or use of glucose-lowering therapy.

*****LDL cholesterol concentrations ≥3.62 mmol/L, HDL cholesterol concentrations <1.03 mmol/L, triglyceride concentrations ≥1.69 mmol/L, or use of lipid-lowering medication.

habits, current alcohol intake, regular exercise habits, obesity, elevated BP, diabetes, and dyslipidemia. There were also similar associations between the WBC count and hypertension across subgroups defined by age, sex, smoking, obesity, and dyslipidemia.

A number of previous observational studies reported associations between indicators of inflammation, such as C-reactive protein, interleukin-6, interleukin-8, WBC count, neutrophil count, neutrophil to lymphocyte ratio, and incidence of hypertension [10–13, 19–21]. A prospective cohort study of atomic bomb survivors in Japan reported that an elevation in the WBC and neutrophil counts, which were measured in the 1960's, were clearly related to an increased risk of future development of hypertension [22]. Our study supports findings from previous studies and showed that there was a significant association between the WBC count and the incidence of hypertension in the Japanese general population in the current era.

**Table 2. Risk of hypertension according to tertiles of the white blood cell count.**

| | White blood cell count | | | |
|---|---|---|---|---|
| | <4,700/μL (N = 951) | 4,700–5,999/μL (N = 1040) | ≥6,000/μL (N = 944) | p value for trend |
| N of events / person-years | 273/4353 | 332/4736 | 303/4097 | |
| Annual incidence rate | 6.3% | 7.0% | 7.4% | |
| Crude hazard ratio (95% confidence interval) | 1.00 (Reference) | 1.12 (0.95–1.31) | 1.17 (1.00–1.38) | 0.056 |
| Adjusted hazard ratio* (95% confidence interval) | 1.00 (Reference) | 1.07 (0.90–1.26) | 1.27 (1.06–1.51) | 0.009 |

*Adjusted for age, sex, smoking, current daily alcohol intake, exercise, obesity, elevated blood pressure, diabetes mellitus, and dyslipidemia.

**Table 3. Multivariable-adjusted hazard ratios of the white blood cell count for the incidence of hypertension in subgroups.**

| | White blood cell count | | | p value for interaction |
|---|---|---|---|---|
| | <4,700/μL (N = 951) | 4,700–5,999/μL (N = 1040) | ≥6,000/μL (N = 944) | |
| Age | | | | |
| <65 years | 1 (reference) | 1.09 (0.88–1.36) | 1.20 (0.96–1.51) | 0.109 |
| ≥65 years | 1 (reference) | 1.01 (0.78–1.30) | 1.08 (0.81–1.44) | |
| Gender | | | | |
| Male | 1 (reference) | 1.09 (0.83–1.43) | 1.07 (0.80–1.42) | 0.775 |
| Female | 1 (reference) | 0.98 (0.79–1.21) | 1.10 (0.87–1.38) | |
| Current smoking | | | | |
| Yes | 1 (reference) | 0.74 (0.44–1.24) | 0.86 (0.54–1.38) | 0.371 |
| No | 1 (reference) | 1.09 (0.92–1.31) | 1.34 (1.10–1.62) | |
| Obesity | | | | |
| Yes | 1 (reference) | 1.28 (0.93–1.75) | 1.34 (0.96–1.86) | 0.577 |
| No | 1 (reference) | 0.99 (0.81–1.20) | 1.22 (0.99–1.52) | |
| Dyslipidemia | | | | |
| Yes | 1 (reference) | 1.00 (0.78–1.29) | 1.21 (0.93–1.57) | 0.889 |
| No | 1 (reference) | 1.12 (0.90–1.40) | 1.29 (1.01–1.65) | |

Values are hazard ratios (95% confidence intervals) adjusted for age (except for subgroup analysis by age), sex (except for subgroup analysis by sex), current smoking (except for subgroup analysis by current smoking), current alcohol intake, regular exercise, obesity (except for subgroup analysis by obesity), elevated blood pressure, and diabetes and dyslipidemia (except for subgroup analysis by dyslipidemia). Obesity: body mass index ≥25 kg/m$^2$. Dyslipidemia: LDL cholesterol concentrations ≥3.62 mmol/l, HDL cholesterol concentrations <1.03 mmol/l, triglyceride concentrations ≥1.69 mmol/l, or use of lipid-lowering medications.

The mechanisms underlying the relationship between WBC count and hypertension has not been clearly defined. An increase in the number of WBCs in blood may promote leukocyte–endothelial cell adhesion, which plays a major role in development and progression of atherosclerosis [14, 19, 23, 24]. Therefore, long-term exposure to increased numbers of WBCs in blood may result in an increase in arterial stiffness and subsequent development of systemic hypertension. Another possibility is that WBC count may in part be associated with hypertension in part as a maker of obesity, because obesity is associated with higher WBC count [25] and weight gain [26] has been shown to be associated with increasing levels in blood pressure levels.

Although this was a large-scale study of the general Japanese population, it has some limitations. First, because of the retrospective nature of the study design, the findings of the present analysis may have been affected by selection bias. Second, people who were aware of healthy behavior were more likely to have attended the health check-ups and to have been included in the present analysis than those with an unhealthy lifestyle. Third, because frequency of people aged 65 years or older ("aging rate") was higher in the Iki City (35.5% in 2015) than that of the total Japan (26.5%in 2015), our findings may not be applicable to other regions of Japan with lower aging rates. Fourth, the actual data of the new onset of hypertension was uncertain because some participants did not return to follow up to health check-up regularly. Fifth, although some studies have reported the relationship of the neutrophil count or neutrophil to lymphocyte ratio with hypertension, we do not have information of the WBC fraction.

## Conclusion

In conclusion, WBC count was associated with future development of hypertension in the general Japanese population. A high-risk strategy under guidance of the WBC count may provide better prevention of future development of hypertension.

## Supporting information

**S1 Fig. Hypertension incidence by tertile of WBC count.**
(DOCX)

**S2 Fig. Risk of developing hypertension by tertile of WBC count.**
(DOCX)

## Acknowledgments

We thank Ellen Knapp, PhD, from Edanz Group (www.edanzediting.com/ac) for editing a draft of this manuscript.

## Author Contributions

**Conceptualization:** Kenji Ito, Tetsuhiko Yasuno, Kosuke Masutani, Hitoshi Nakashima.

**Data curation:** Kosuke Masutani, Hitoshi Nakashima.

**Formal analysis:** Shintaro Ishida, Shunsuke Funakoshi, Atsushi Satoh, Toshiki Maeda, Miki Kawazoe, Chikara Yoshimura, Kazuhiro Tada, Koji Takahashi.

**Funding acquisition:** Kosuke Masutani.

**Investigation:** Shunsuke Funakoshi, Atsushi Satoh, Toshiki Maeda, Miki Kawazoe, Chikara Yoshimura, Kazuhiro Tada, Koji Takahashi, Kosuke Masutani.

**Methodology:** Toshiki Maeda, Miki Kawazoe, Chikara Yoshimura, Kazuhiro Tada, Koji Takahashi, Tetsuhiko Yasuno, Kosuke Masutani.

**Project administration:** Kenji Ito, Tetsuhiko Yasuno, Hitoshi Nakashima.

**Resources:** Kenji Ito, Tetsuhiko Yasuno, Kosuke Masutani, Hitoshi Nakashima.

**Supervision:** Seiji Kondo, Chikara Yoshimura, Hisatomi Arima.

**Writing – original draft:** Shintaro Ishida, Hisatomi Arima.

**Writing – review & editing:** Hisatomi Arima.

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
