## [Decision Letter · Decision Letter 0]

6 Oct 2020

PONE-D-20-20652

White blood cell count and incidence of hypertension in the general Japanese population: ISSA-CKD study

PLOS ONE

Dear Dr. Ishida,

Thank you for submitting your manuscript to PLOS ONE. After careful consideration, we feel that it has merit but does not fully meet PLOS ONE’s publication criteria as it currently stands. Therefore, we invite you to submit a revised version of the manuscript that addresses the points raised during the review process.

We look forward to receiving your revised manuscript.

Kind regards,

Tatsuo Shimosawa, M.D., Ph.D.

Academic Editor

PLOS ONE

Journal Requirements:

2.Thank you for stating the following in the Financial Disclosure  section:

[HA received research grants from Daiichi Sankyo and Takeda, lecture fees from Bayer, Daiichi Sankyo, Fukuda Denshi, MSD, Takeda, and Teijin, and fees for consultancy from Kyowa Kirin. There are no other conflicts of interest to disclose.]. 

We note that you received funding from a commercial source: [Daiichi Sankyo,Takeda, Bayer, Fukuda Denshi, MSD, Teijin and Kyowa Kirin]

Reviewers' comments:

Reviewer's Responses to Questions

**Comments to the Author**

1. Is the manuscript technically sound, and do the data support the conclusions?

Reviewer #1: Yes

Reviewer #2: Yes

2. Has the statistical analysis been performed appropriately and rigorously? 

Reviewer #1: No

Reviewer #2: Yes

3. Have the authors made all data underlying the findings in their manuscript fully available?

Reviewer #1: Yes

Reviewer #2: Yes

4. Is the manuscript presented in an intelligible fashion and written in standard English?

Reviewer #1: Yes

Reviewer #2: Yes

5. Review Comments to the Author

Reviewer #1: Although it is a retrospective analysis, it is an interesting study that analyzes the association between the development of hypertension and white blood cell(WBC) count in many subjects. However, more detail discussion should be given to the causal relationship between WBC count and hypertension. For example, if it is reported that the number of WBC decreases due to improvement of obesity and lipid abnormalities, it becomes easier to support the results of this study. Finally, it is necessary to correct LDH cholesterol to LDL cholesterol in Table 2.

Reviewer #2: Using a population-based retrospective cohort study in Iki-City, Nagasaki Prefecture, Japan, authors evaluated the relationship between the WBC count and hypertension in the general Japanese population. They demonstrated that the WBC count was associated with future development of hypertension.

It should be recommended to add some explanations and/or interpretation.

1. Iki-City is considered to be an aging society compared to the national average in Japan. It may affect as selection bias.

2. A number of previous observational studies reported associated between indicators of inflammation and incidence of hypertension. A prospective cohort study of a Japanese population was also reported. What is the novelty in the present study?

6. PLOS authors have the option to publish the peer review history of their article (what does this mean?). If published, this will include your full peer review and any attached files.

Reviewer #1: No

Reviewer #2: No

---

## [Author Response · Author response to Decision Letter 0]

8 Jan 2021

Response to Reviewer #1 

Thank you for your useful suggestions. We have attempted to address your suggestions as follows: 

Major comment 1. Although it is a retrospective analysis, it is an interesting study that analyzes the association between the development of hypertension and white blood cell (WBC) count in many subjects. However, more detail discussion should be given to the causal relationship between WBC count and hypertension. For example, if it is reported that the number of WBC decreases due to improvement of obesity and lipid abnormalities, it becomes easier to support the results of this study.

Response. Thank you for your advice. (1) We could not find a paper that reported that WBC decreased due to improvement of obesity but there was a cross sectional study that reported the association between obesity and higher WBC counts. Because weight gain has been shown to be associated with increase in blood pressure levels, WBC counts may in part be associated with hypertension in part as a maker of obesity. We have added this the 3rd paragraph of the Discussion section (lines 167 to 170, page 8) as follows:

“Another possibility is that WBC count may in part be associated with hypertension in part as a maker of obesity, because obesity is associated with higher WBC count [25] and weight gain [26] has been shown to be associated with increasing levels in blood pressure levels. “

(2) We could find papers which reported that WBC decreased due to lipid lowering treatment using statins. However, because statins have anti-inflammatory effects, it is not clear whether lowering lipid abnormalities lowering itself reduces WBC counts or not. Therefore, we are not sure whether lipids are involved in the link between WBC counts and hypertension. 

Major comment 2. It is necessary to correct LDH cholesterol to LDL cholesterol in Table 2.

Response. Thank you for your advice. We suppose that the typo you pointed out is in Table1 instead of Table2. We corrected “LDH cholesterol” to “LDL cholesterol” in the revised Table1.

Response to Reviewer #2 

Thank you for your useful suggestions. We have attempted to address your suggestions as follows: 

Major comment 1. Iki-City is considered to be an aging society compared to the national average in Japan. It may affect as selection bias.

Response. As you pointed out, frequency of people aged 65 years or older (“aging rate”) was 35.5% in the Iki City in 2015, which was higher than that of the total Japan (26.5%in 2015). As a result, findings of this study with higher frequency of elderly may not be applicable to other regions of Japan with lower aging rates. We have added this limitation to the 4th paragraph of the Discussion section (lines 176 to 178, page 8) as follows:

“Third, because frequency of people aged 65 years or older (“aging rate”) was higher in the Iki City (35.5% in 2015) than that of the total Japan (26.5%in 2015), our findings may not be applicable to other regions of Japan with lower aging rates.”

Major comment 2. A number of previous observational studies reported associated between indicators of inflammation and incidence of hypertension. A prospective cohort study of a Japanese population was also reported. What is the novelty in the present study?

Response. Thank you for your advice. A novelty of the present study is the fact that this is a population-based study in current Japan. Although one longitudinal study investigated the association between white blood cell counts and incidence of hypertension among Japanese, their participants were atomic bomb survivors in the Hiroshima and the Nagasaki Prefectures. Furthermore, baseline assessment of white blood cells counts were conducted in 1960’s. In contrast, participants of this study were general Japanese people and baseline assessment was conducted after 2006. 

Yours sincerely 

Shintaro Ishida

---

## [Decision Letter · Decision Letter 1]

18 Jan 2021

White blood cell count and incidence of hypertension in the general Japanese population: ISSA-CKD study

PONE-D-20-20652R1

Dear Dr. Ishida,

We’re pleased to inform you that your manuscript has been judged scientifically suitable for publication and will be formally accepted for publication once it meets all outstanding technical requirements.

Kind regards,

Tatsuo Shimosawa, M.D., Ph.D.

Academic Editor

PLOS ONE

Additional Editor Comments (optional):

Reviewers' comments:

Reviewer's Responses to Questions

**Comments to the Author**

1. If the authors have adequately addressed your comments raised in a previous round of review and you feel that this manuscript is now acceptable for publication, you may indicate that here to bypass the “Comments to the Author” section, enter your conflict of interest statement in the “Confidential to Editor” section, and submit your "Accept" recommendation.

Reviewer #1: (No Response)

Reviewer #2: All comments have been addressed

2. Is the manuscript technically sound, and do the data support the conclusions?

Reviewer #1: Yes

Reviewer #2: Yes

3. Has the statistical analysis been performed appropriately and rigorously? 

Reviewer #1: Yes

Reviewer #2: Yes

4. Have the authors made all data underlying the findings in their manuscript fully available?

Reviewer #1: Yes

Reviewer #2: Yes

5. Is the manuscript presented in an intelligible fashion and written in standard English?

Reviewer #1: Yes

Reviewer #2: Yes

6. Review Comments to the Author

Reviewer #1: I confirmed the detail discussion about causal relationship between WBC count and hypertension. I was satisfied with your response and revised manuscript. I decided to accept this paper.

Reviewer #2: This second version of the paper has been much improved. The responses to the comments and the added description are adequate.

7. PLOS authors have the option to publish the peer review history of their article (what does this mean?). If published, this will include your full peer review and any attached files.

Reviewer #1: No

Reviewer #2: No

---

## [Editor Report · Acceptance letter]

22 Jan 2021

PONE-D-20-20652R1 

White blood cell count and incidence of hypertension in the general Japanese population: ISSA-CKD study 

Dear Dr. Ishida:

I'm pleased to inform you that your manuscript has been deemed suitable for publication in PLOS ONE. Congratulations! Your manuscript is now with our production department. 

Kind regards, 

on behalf of

Prof. Tatsuo Shimosawa 

Academic Editor

PLOS ONE